# Spanish Grammatical Gender Interference in Papiamentu

**Jorge R. Valdés Kroff [1,\*], Frederieke Rooijakkers [2] and M. Carmen Parafita Couto [2,\*]** 

[1]  Spanish and Portuguese Studies, University of Florida, Gainesville, FL 32611, USA
[2]  Centre for Linguistics, Leiden University, 2311 EZ Leiden, The Netherlands;
    frederiekerooijakkers@hotmail.nl
\*  Correspondence: jvaldeskroff@ufl.edu (J.R.V.K.); m.parafita.couto@hum.leidenuniv.nl (M.C.P.C.);
    Tel.: +1-352-273-3744 (J.R.V.K.); +31-71-527-2644 (M.C.P.C.)

**Abstract:** The aim of this study is to determine whether Spanish-like gender agreement causes interference in speakers of Papiamentu (a Western Romance-lexified creole language) who also speak Spanish. Papiamentu and Spanish are highly cognate languages in terms of their lexicons. However, Papiamentu lacks grammatical gender assignment and agreement, leading to cognate words with major morpho-syntactic differences. A total of 41 participants with different linguistic profiles (Papiamentu-dominant, Dutch-dominant, Spanish-dominant, and Spanish heritage speaker-Papiamentu bilinguals) listened to 82 Papiamentu sentences, of which 40 contained a Spanish-like gender-agreeing element on the Determiner, Adjective, or Determiner + Adjective and with half of the experimental items marked with overtly masculine (i.e., *-o*) or feminine (i.e., *-a*) gender morphology. Participants performed a forced-choice acceptability task and were asked to repeat each sentence. Results showed that Spanish-dominant speakers experienced the greatest interference of Spanish gender features in Papiamentu. This suggests that in cases where speakers must suppress gender in their second language (L2), this is not easy to do. This is especially the case in highly cognate languages that differ in whether they realize gender features.

**Keywords:** grammatical gender; interference; cognates; Papiamentu; Spanish

## 1. Introduction

Grammatical differences between languages often result in difficulties in second language (L2) acquisition and bilingual strategies in language contact (e.g., Hopp 2013; Muysken 2013). This cross-linguistic interference is likely to occur in the case of highly cognate languages due to increased lexical and semantic overlap. The current study examines what happens when speakers of a morphologically rich language (Spanish) also speak a highly cognate language that lacks a morphological feature (Papiamentu). We examine this issue through gender agreement due to it being a well-studied morpho-syntactic element in first language (L1) and L2 acquisition (Montrul 2004). Its difficulty for L2 speakers who speak languages that lack gender agreement is also well-documented (e.g., Eddington 2002; Grüter et al. 2012; Montrul et al. 2008).

Papiamentu is an Iberian-lexifier creole language spoken on the islands of Aruba, Bonaire, and Curaçao (former Dutch Antilles) and in the Netherlands. The total number of Papiamentu speakers is approximately 200,000. Papiamentu speakers in Curaçao are highly multilingual, often speaking Papiamentu, Dutch, English, and Spanish to varying degrees. Despite the variety of languages present on the islands, Papiamentu is the first language of more than 80% of the population on the Caribbean islands (Kester 2011).

### 1.1. The Nominal Domain in Spanish and Papiamentu

Grammatically speaking, Spanish nouns are either masculine or feminine. Although the gender distribution between masculine and feminine nouns is roughly half, masculine is characterized as the default or unmarked gender (Harris 1991). Most Spanish nouns and adjectives mark grammatical gender in canonical endings such as *-o* for masculine and *-a* for feminine. Spanish determiners and adjectives agree with the noun in gender and number, with most adjectives following the noun. Unlike Spanish, Papiamentu has no gender distinction but the relative position of Papiamentu adjectives with respect to the noun is like Spanish. Adjectives in Papiamentu are invariant typically ending in *-o* or *-u*. Examples (1) and (2) illustrate the lexical and word order similarities between Spanish (a) and Papiamentu (b).

1. a. $La_{fem}$ $mesa_{fem}$ $redonda_{fem}$
   b. $E_{\emptyset}$ $mesa_{\emptyset}$ $rondó_{\emptyset}$
      The table round
      'The round table'
2. a. $El_{masc}$ $pato_{masc}$ $blanco_{masc}$
   b. $E_{\emptyset}$ $patu_{\emptyset}$ $blanku_{\emptyset}$
      The duck white
      'The white duck'

### 1.2. Gender Interference in Palenquero-Spanish Speakers

Lipski (2015) investigated whether Spanish-Palenquero bilinguals accept and/or reproduce Spanish gender agreement in Palenquero. Palenquero is a Spanish-based Afro-Colombian creole language spoken in San Basilio de Palenque, Colombia. Palenquero and Spanish share largely cognate lexicons while Palenquero morpho-syntax is what Lipski describes as a subset of Spanish (i.e., it lacks grammatical gender).

To determine whether Spanish-like feminine gender agreement could be observed in Palenquero, Lipski (2015) first used a picture-describing task. He tests 10 first language (L1) Palenquero speakers, 10 Palenquero heritage speakers, 10 L2 Palenquero speakers, and 4 Palenquero language instruction teachers. The results confirm his hypotheses that Spanish gender agreement cannot be fully suppressed by L2 Palenquero speakers, who introduce some Spanish-like feminine gender agreement in Palenquero determiners and adjectives modifying nouns whose Spanish cognates are grammatically feminine. In contrast, L1 speakers and the metalinguistically sensitive Palenquero language teachers exhibit little Spanish-like feminine agreement. Nonetheless, the high activation level of Spanish prompted by the large number of cognates results in some carryover of gender agreement. Heritage speakers show greater inter-speaker variability.

Lipski additionally utilizes an acceptability task in which participants listen to stimuli and state whether the utterance is "good" Palenquero or not. Afterwards, they repeat each sentence exactly as they have heard it, regardless of their own intuitions on the acceptability of the sentence. Here, 12 L1 Palenquero speakers, 12 heritage Palenquero speakers, 15 L2 Palenquero speakers, and 6 Palenquero language teachers are tested. Results show that L1 speakers and teachers pattern together in accepting about half of the feminine gender-agreement stimuli. Heritage and L2 speakers, on the other hand, display an acceptance level of around 75%. As Lipski predicted, L1 speakers change many feminine endings in *-a* to the well-formed Palenquero gender-invariant *-o* while L2 speakers rarely do, thus demonstrating Spanish interference due to acceptance of Spanish-like gender agreement. Palenquero language teachers behave like L1 speakers when modifiers are immediately adjacent to the head noun but more like heritage speakers for predicate adjectives. Lipski suggests that even though the teachers are metalinguistically aware, Palenquero is not their dominant language.

In a more recent study, Lipski (2017) examines the tradeoff between the on-line construction of modifier-noun gender agreement and the automatization of agreement. In this study, he focuses on L1 Spanish speakers who are acquiring L2 Palenquero. When switching from the gender-agreeing L1

to the gender-less L2, the persistence or absence of gender agreement in cognate items is an indirect measure of the cost differential between producing morpho-syntactic agreement and suppressing the carryover of obligatory agreement to the L2. To test this, Lipski conducts a number recall + repetition experiment and a speeded translation task. The results reveal the strong influence of L1 Spanish gender agreement on L2 Palenquero. Furthermore, heritage Palenquero speakers' retention of gender agreement falls between L1 and L2 speakers.

Taken together, the results of Lipski's (2015, 2017) studies suggest that "less" is not always preferred to "more". Lipski posits that the appearance of Spanish-like gender agreement in the L2 and heritage Palenquero speakers may be due to the failure to inhibit cognate Spanish items and the corresponding syntactic projections responsible for gender agreement.

*1.3. Current Study: Spanish Grammatical Gender Interference in Papiamentu*

The current study extends Lipski (2015, 2017) to Papiamentu-Spanish multilinguals[1]. We test if dominant Papiamentu speakers—speakers born into Papiamentu-speaking families and raised on the islands who are also exposed to Spanish—accept and/or reproduce Spanish-like gender agreement in Papiamentu. Following Lipski, we hypothesize that dominant Papiamentu speakers will reject Spanish gender agreement in Papiamentu sentences. In contrast, Spanish heritage speakers, who have been raised and schooled in a Papiamentu-majority environment but whose home language is Spanish, and Spanish-dominant L2 speakers of Papiamentu who immigrated to the islands after puberty, are predicted to accept more sentences that contain Spanish-like gender agreement.

We additionally include a group of dominant Dutch speakers whose families immigrated from the Netherlands but who are brought up on the islands (and who are also exposed to Spanish) to test whether the presence of grammatical gender in the L1 more generally may lead to gender agreement interference. Like Spanish, Dutch has a two-way gender system which distinguishes between common gender (nouns that are preceded by the Dutch article *de*) and neuter gender (nouns that are preceded by the Dutch article *het*). These two gender categories are distributed unequally as the common gender comprises around 75% of all Dutch nouns (Pablos et al. 2019). The Dutch gender system is more opaque, and the distinction between common and neuter gender in Dutch is neutralized in the plural form (preceded by the Dutch article *de*).

Despite the experimental design and linguistic similarities between the two language pairs, our study is different from Lipski's (2015) study in several ways. First, most participants in Lipski's experiments were dominant Spanish speakers acquiring L2 Palenquero in a sociopolitical context in which Spanish is the prestige, government-sanctioned language and in which formal education is conducted. In our Papiamentu-Spanish study, Spanish remains primarily a minority language with environmental presence on the media and through tourism and is supplemented with formal education in public schools from eighth grade in Curaçao, partly due to close geographic proximity to Venezuela. Thus, this study allows us to examine the directionality of cross-linguistic effects of morpho-syntactic transfer in cognate languages and to compare the role of environmental factors.

## 2. Materials and Methods

*2.1. Participants*

Forty-one participants were tested in Curaçao during the period of June–August 2018. The participants were divided into four different categories, illustrated in (3):

---

[1]　We refer to our participants as multilinguals because the residents on the island are regularly exposed to Dutch, English, Papiamentu, and Spanish to varying degrees, with the dominant or preferred language often intersecting with racial identity (Kester 2011). For further group characteristics and an extended discussion on Dutch grammatical gender, we refer the reader to the text and Table S1 in the Supplementary Materials.

3.  a.    Dutch Dominant (n = 7)
    b.    Papiamentu Dominant (n = 22)
    c.    Spanish Dominant (n = 6)
    d.    Heritage Spanish (HS) Papiamentu (n = 6)

This group division was determined by the responses of a linguistic background questionnaire based on (1) self-reported Spanish and Papiamentu proficiency, (2) the age of acquisition of Spanish and Papiamentu, (3) the language spoken at home and the language spoken at school, and (4) the country of birth (see Table S1 in Supplementary Materials). Participants in the Dutch-dominant group were born in Curaçao, learned Papiamentu at a young age, and speak Dutch at home as the dominant language. Additionally, most participants in this group lived in the Netherlands when the experiment took place and thus are primarily exposed to Dutch. The Papiamentu-dominant group consists of L1 Papiamentu speakers that learned Papiamentu at a young age and grew up in households where Papiamentu was the dominant language. Most of the participants in this group are multilingual and learned Spanish in primary school. Two participants in this group were exposed to Spanish under the age of two. For the Spanish-dominant group, participants were born in a Spanish-speaking country, speak Spanish at home, and were exposed to Papiamentu as adults. Finally, the Spanish HS-Papiamentu group includes participants who were either born in a Spanish-speaking country or in Curaçao and learned Papiamentu in primary school. Their home language was reported to be Spanish. All participants of this group moved to Curaçao at a young age and still live in Curaçao today.

*2.2. Materials*

Eighty-two Papiamentu sentences were created, of which 38[2] contained a Spanish-like gender-agreeing element either on the determiner (n = 8), the adjective (n = 12), or both (n = 18). Out of these 38 manipulated sentences, 18 contained combinations of adjectives and/or determiners whose Spanish cognates are feminine (e.g., adjectives and/or determiners ending in –*a*). The other twenty sentences contained Spanish-like masculine gender-agreement (e.g., adjectives or plural determiners ending in –*o(s)*). The remaining 42 filler sentences were Papiamentu sentences with no gender manipulations (i.e., "correct" Papiamentu sentences). Two native Papiamentu speakers from Curaçao verified if the manipulated stimuli were correct Papiamentu sentences, apart from the experimentally manipulated target determiners and/or adjectives. All sentences were recorded by a Papiamentu-Spanish male speaker. After the recordings, the items were shortened with PRAAT software (version 5.3.16; Boersma and Weenink 2012). The entire list of 82 stimuli was randomized in Excel using the (=RAND) function, and four different lists were created. All stimuli were loaded on a laptop, and headphones with a built-in microphone were used to record responses. Two examples of stimuli that were used in the experiment, containing a masculine and feminine Spanish gender-agreeing element, are provided below in (5):

5.  a.    **Spanish-like feminine gender agreement**[3]:

| | | | | |
|---|---|---|---|---|
| Example stimulus: | $La_{fem}$ pluma | $blanka_{fem}$ ta | suave | |
| Papiamentu Equivalent: | E    pluma | blanku    ta | suave | |
| | det  feather | white    TAM | soft | |
| | 'The white feather is soft' | | | |

   b.    **Spanish-like masculine gender agreement:**

| | | | | |
|---|---|---|---|---|
| Example stimulus: | E    paranan | $chikito_{masc}$ ta | kanta $bunito$[4] | |
| Papiamentu Equivalent: | E    paranan | chikitu    ta | kanta bunita | |
| | det  bird-PL | small    TAM | sing   beautiful | |
| | 'The small birds are singing beautifully' | | | |

---

2    The original dataset had 40 experimental stimuli, but we subsequently discovered incorrect coding on 2 sentences and removed these from analyses.

As shown in (5), the manipulated adjectives and determiners occurred in different positions in the sentence (i.e., post-nominal and predicate adjectives were included). Tables 1 and 2 provide a more detailed overview of the distribution of the gendered adjectives and determiners, respectively.

**Table 1.** Examples of experimentally-manipulated adjectives.

| Adjective Endings | Feminine *-a* (*-o* in PAP [1]) | Feminine *-a* (*-u* in PAP) | Masculine *-o* (*-a* in PAP) | Masculine *-o* (*-u* in PAP) |
|---|---|---|---|---|
| Example stimulus | rondá | chikita | delegó | blanko |
| Papiamentu | rondó | chikitu | delegá | blanku |
| Spanish equivalent | redondo/a | pequeño/a | delgado/a | blanco/a |
| English translation | 'round' | 'small' | 'thin' | 'white' |

[1] PAP = Papiamentu.

**Table 2.** Examples of experimentally-manipulated determiners.

| Determiners | Masculine Singular *el* [1] (*e* in PAP) | Feminine Singular *la/una* (*e/un* in PAP) | Masculine Plural *los* (*e* in PAP) | Feminine Plural *las* (*e* in PAP) |
|---|---|---|---|---|
| Example stimulus | **el** aros | **una** kara | **los** piskánan | **las** islanan |
| Papiamentu | e aros | un kara | e piskánan | e islanan |
| Spanish equivalent | el arroz | una cara | los peces/pescados | las islas |
| English translation | 'the rice' | 'a face' | 'the fish (plural)' | 'the islands' |

[1] Spanish indefinite determiner *un* was not used because of its cognate status with Papiamentu.

### 2.3. Procedure

The experiment followed the Ethics Code for linguistic research in the faculty of Humanities at Leiden University, which approved its implementation. Participants were instructed that they would listen to 82 Papiamentu sentences over noise-cancelling headphones. Each sentence was immediately followed by a short "beep" sound. Upon hearing the beep, all participants were asked to indicate if the sentence was correct Papiamentu[5] by responding with "yes" or "no" within two seconds (i.e., acceptability judgment task) and to repeat the Papiamentu sentence exactly as they heard it (i.e., sentence repetition task). All answers outside of this time window were not used for the analysis, and the instructions were given in English or Dutch (i.e., languages in which the second author could provide instructions). All 41 participants completed the task without any objection and all answers were digitally recorded. After completing both tasks, all participants completed a language history questionnaire and signed a consent form giving permission to use all recorded data. Participants had the option to complete all forms in Spanish or Papiamentu (languages in which all participants were literate).

### 3. Results

We report on the results for accuracy for the acceptability judgment task: for experimental trials, the expected response is 'no'. First, we analyzed the unchanged Papiamentu filler items in which the expected response is 'yes' to ensure that participants were not randomly selecting answers. One participant from the Papiamentu-dominant group was removed from this analysis and all subsequent analyses for having scored only about half correct (45%) on filler trials. For the remaining 40 participants,

---

[3]    For glosses, det = determiner, TAM = tense-aspect-mood particle, and PL = plural.

[4]    A reviewer rightly points out that *bonito* is an adverb derived from an adjective and would not show gender agreement in Spanish. This is the only experimental sentence with such characteristics. For a list of experimental sentences, see Appendix A.

[5]    For this experiment, 'correct' Papiamentu means that the participants would consider the Papiamentu sentence to be a grammatically well-formed sentence when speaking to another Papiamentu speaker.

the Dutch-dominant group correctly identified 93% (range: 76–100%); the Papiamentu-dominant group correctly identified 90% (range: 73–100%); the Spanish-dominant group correctly identified 89% (range: 80–95%); and the Heritage Spanish-Papiamentu group correctly identified 88% (range: 88–100%) of filler items. Thus, all remaining participants show high accuracy on identifying correct Papiamentu sentences.

For the main analysis, we conducted a 3 × 2 × 4 repeated-measures ANOVA in R (v. 3.5.1) in which the dependent variable is the proportion of correctly identifying experimental items as not well-formed, with the within-subjects factors Condition (Adjective, Determiner, Determiner + Adjective) and Gender (Masculine, Feminine) and the between-subjects factor Group (Dutch-dominant, Papiamentu-dominant, Spanish-dominant, Heritage Spanish-Papiamentu). The dataset consisted of 1558 tokens out of a possible 1640 tokens. The omnibus model revealed a main effect for Condition $(F[2,72] = 10.17, p < 0.001)$, a main effect for Gender $(F[1,36] = 20.37, p < 0.001)$, and a main effect for Group $(F[3, 36] = 29.29, p < 0.001)$. The model also confirmed an interaction between Group and Condition $(F[6,72] = 5.79, p < 0.001)$ and a 3-way Condition x Gender x Group interaction $(F[6,72] = 5.44, p < 0.001)$. Due to the 3-way interaction, we conducted separate 3 × 2 repeated-measures ANOVAs per group.

## 3.1. Dutch-Dominant Group

For the Dutch-dominant group (n = 7), the statistical model revealed a main effect for Condition $(F[2,12] = 5.4, p = 0.021)$ and a main effect for Gender $(F[1,6] = 12.43, p = 0.012)$. There was no significant interaction between the two variables. As illustrated in Figure 1, this group was least accurate with the determiner condition and least accurate on masculine-marked trials.

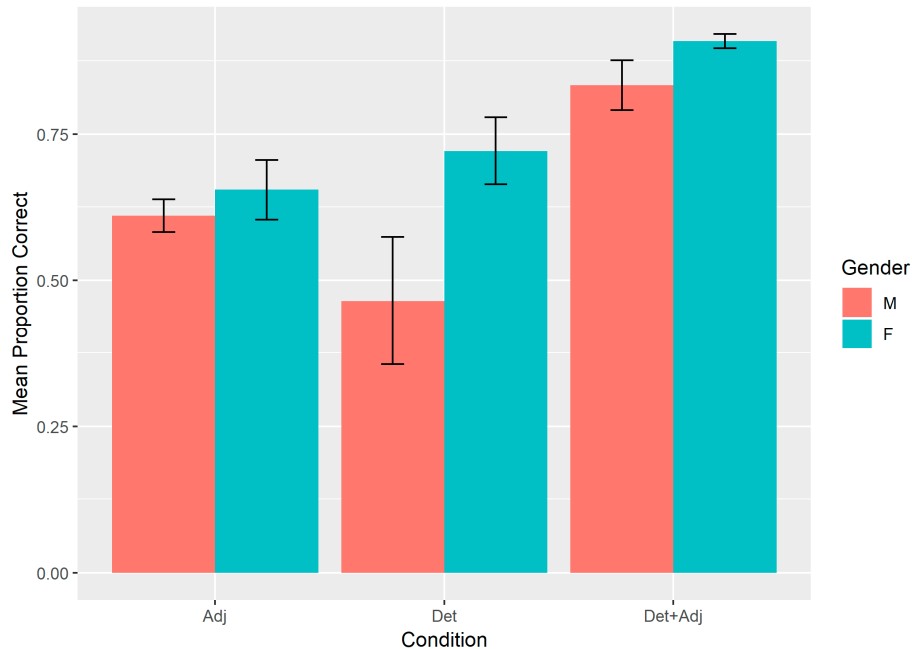

**Figure 1.** Mean proportion accuracy for the Dutch-dominant group. Error bars represent +/− 1 Standard Error of the Mean. Condition is plotted on the horizontal axis. Adj = adjective, Det = determiner, Det + Adj = determiner + adjective, M = masculine, F = feminine.

## 3.2. L1 Papiamentu-Dominant Group

For the Papiamentu-dominant group (n = 21), the statistical model revealed a main effect for Condition $(F[2,40] = 11.83, p < 0.001)$, a main effect for Gender $(F[1, 20] = 19.14, p < 0.001)$, and a significant interaction between Condition and Gender $(F[2,40] = 3.86, p = 0.029)$. Due to the interaction, we conducted pairwise comparisons corrected for multiple comparisons using Tukey's test. In

comparisons that test differences between gender within the same condition (e.g., feminine-marked vs. masculine-marked determiners, adjectives, or determiners + adjectives), the difference between feminine- and masculine-marked adjectives was significant (difference = 0.18, t = 4.639, $p < 0.001$), indicating that this group was more accurate on correctly rejecting trials in which the adjective was overtly marked with Spanish-like feminine agreement. Among contrasts of the same gender type but across conditions, the difference between masculine-marked determiners and adjectives was significant (difference = 0.16, t = 4.128, $p = 0.001$) as well as the difference between masculine-marked Determiner + Adjective trials and masculine-marked adjectives (difference = 0.174, t = 4.478, $p = 0.001$). In both cases, the rejection of Spanish-like masculine-marked adjectives was less accurate than the other conditions. All other contrasts were not significant (ps > 0.19). The results are plotted in Figure 2.

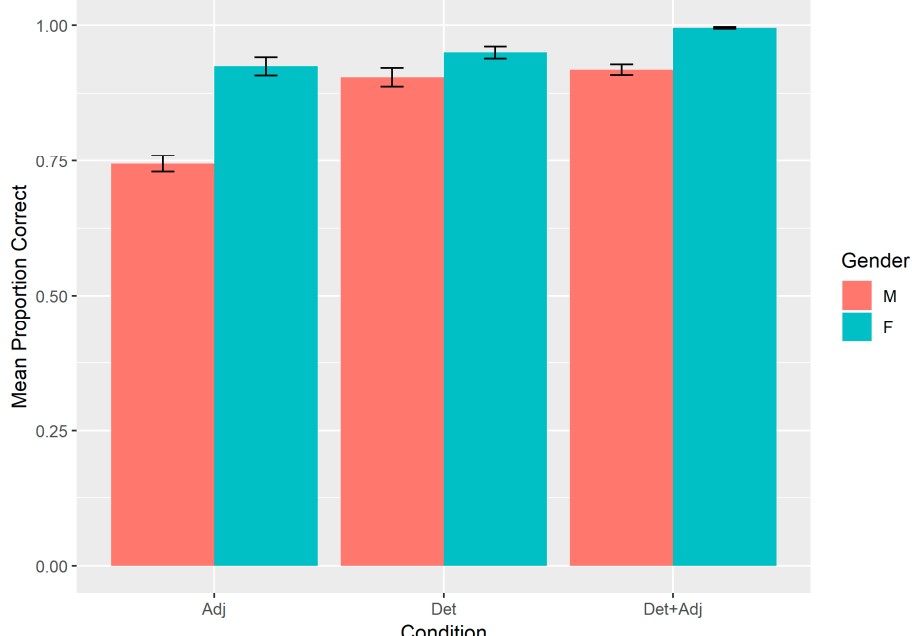

**Figure 2.** Mean proportion accuracy for the Papiamentu-dominant group. Error bars represent +/− 1 Standard Error of the Mean. Condition is plotted on the horizontal axis. Adj = adjective, Det = determiner, Det + Adj = determiner + adjective, M = masculine, F = feminine.

### 3.3. Spanish-Dominant Group

For the Spanish-dominant group (n = 6), the model revealed a main effect for Condition (F[2,10] = 4.147, $p = 0.049$) and a significant interaction between Condition and Gender (F[2,10] = 18.296, $p < 0.001$). No main effect was detected for Gender. We again conducted pairwise comparisons using Tukey's test. Only the contrast between masculine-marked determiners and adjectives was significant (difference = −0.494, t = −3.25, $p = 0.031$). This contrast indicates that the Spanish-dominant group was less accurate in rejecting Spanish-like masculine-marked features when manipulated on the determiner and more accurate with masculine-marked adjectives. All other contrasts were not significant (ps > 0.35). Results are plotted in Figure 3.

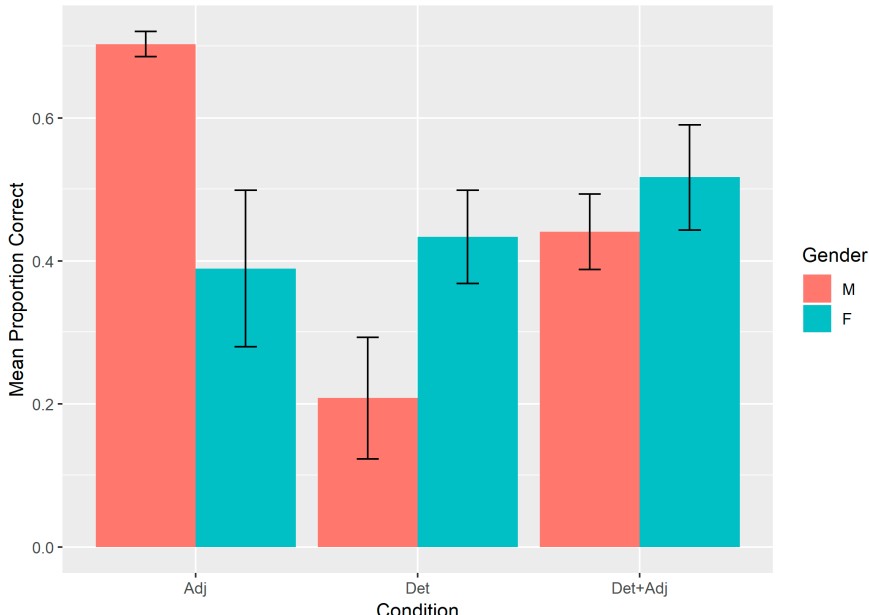

**Figure 3.** Mean proportion accuracy for the Spanish-dominant group. Error bars represent +/− 1 Standard Error of the Mean. Condition is plotted on the horizontal axis. Adj = adjective, Det = determiner, Det + Adj = determiner + adjective, M = masculine, F = feminine.

### 3.4. Heritage Spanish-Papiamentu Group

For the Spanish heritage speaker group (n = 6), the statistical model only found a marginal effect for Gender (F[1,5] = 5.044, $p$ = 0.075) and no main effect for Condition or interaction between Condition and Gender. The marginal effect is reflected on the overall lower accuracy on masculine-marked trials as depicted in Figure 4.

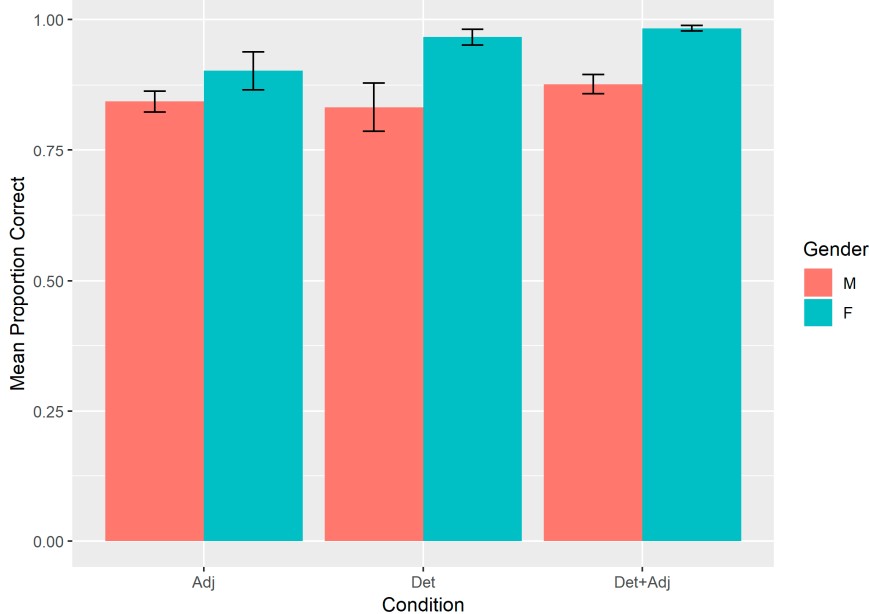

**Figure 4.** Mean proportion accuracy for the Spanish Heritage Spanish (HS)-Papiamentu group. Error bars represent +/− 1 Standard Error of the Mean. Condition is plotted on the horizontal axis. Adj = adjective, Det = determiner, Det + Adj = determiner + adjective, M = masculine, F = feminine.

## 4. Discussion

Most L2 acquisition studies that focus on how grammatical gender is acquired by speakers of a non-gendered language have shown that acquisition of gender assignment and agreement is difficult. Similar to Lipski (2015), we have shown that this difficulty is bidirectional. That is, in cases where the L2 speaker must suppress gender, gender interference can happen. This is especially the case in highly cognate languages that differ in whether they encode grammatical gender. Spanish-dominant speakers experienced the greatest interference of Spanish gender features in Papiamentu. However, the Dutch-dominant group also scored lower in rejecting Spanish gender features as compared to Papiamentu-dominant and Spanish HS-Papiamentu groups, possibly indicating that the presence of gender in Dutch also played a role (Sabourin and Stowe 2008). At the same time, Spanish HSs were better than Spanish-dominant speakers at suppressing gender interference. This suggests that going from a dominant language with gender to a language without gender is harder than suppressing gender from a less-dominant language.

Regarding interference according to word type (determiners and adjectives), the general tendency is to experience more interference with Determiners as compared to Adjectives (Dutch-dominant, Spanish-dominant, Heritage Spanish-Papiamentu). Speculatively, this difference may be due to the salience of determiners vs. adjectives as adjectives are also lexical (i.e., semantic) elements. From a syntactic point of view, determiners typically introduce the phrases in which they appear and have a fixed position. Abney (1987) posited in his "DP hypothesis" that the head of a nominal phrase is a determiner, D, rather than a noun (N). Under this account, determiners do not occupy the Specifier position of the NP. Instead, the determiner is the head of the DP. Attributive adjectives, on the other hand, have a more flexible order and are dependent on the noun, usually being treated as adjuncts.

As for interference related to gender (masculine vs. feminine), the results generally point towards greater interference on words marked with a Spanish masculine feature (*-o*) compared to Spanish feminine (*-a*), the exception being feminine-marked adjectives in the Spanish group. This was not surprising given the status of feminine as marked gender in Spanish and masculine as default (Harris 1991). In other words, the surfacing of feminine-marked morphemes is likely to be more salient and, subsequently, more easily rejected by our participants.

Moving away from the phenomenon under investigation, we see an interesting parallel with the current results and our own prior research on code-switching between gendered and non-gendered languages and the use of the analogical criterion vs. the default gender strategy. Across different language pairs and bilingual communities, the analogical criterion strategy (i.e., transfer of gender assignment to non-gendered language) seems to be absent from speakers who are not Spanish L1 speakers (cf. (Bellamy et al. 2018) for Purepecha-Spanish bilinguals), while L1 Spanish speakers seem more likely to follow the analogical criterion (see (Liceras et al. 2008) for Spanish-English or (Munarriz et al. 2019) for Basque-Spanish). At the same time, certain bilingual communities may also settle on specific code-switching patterns. For example, Valdés Kroff (2016) observed that Spanish-English bilinguals in Miami tend to use masculine as default, and Królikowska et al. (2019) compared the gender assignment patterns of four Spanish-English bilingual populations and observed that the more the bilinguals engaged in code-switching, the greater the tendency to assign the default masculine gender to mixed nominal constructions. Thus, the observed differences in gender assignment strategies across communities and language pairs may be due to a combination of proficiency and environmental factors, which we believe are factors that also play a greater role in the current study and should more explicitly be addressed in future studies.

We especially acknowledge that proficiency in Papiamentu may have played a role in our current results. It is perhaps not surprising that those groups who arguably have the highest proficiency in Papiamentu (Papiamentu-dominant and Heritage Spanish-Papiamentu) are least likely to experience gender interference, although we predicted greater interference for the latter group. Nevertheless, The Dutch-dominant group experienced greater interference despite their arguably weaker proficiency in Spanish. One speculative account for this finding is due to the presence of grammatical gender

in Dutch, even though the system is quite different (see Supplementary Materials). However, both Spanish and Dutch have binary gender systems, and both have default genders (masculine in Spanish, common in Dutch). Finally, one limitation of the current study are the small sample sizes for three of our groups. We are currently collecting data from a population of Papiamentu speakers in the Netherlands (where Spanish is not common) to test whether Spanish gender interference would be reduced. However, this may in turn augment possible Dutch gender interference.

What is noteworthy is that we observe a similar entrenchment effect of L1 Spanish gender across (i) code-switching studies in different bilingual populations, (ii) Lipski's (2015, 2017) studies on Palenquero-Spanish, and (iii) our current study on Papiamentu-Spanish. The state of the research to date calls for further studies to be able to determine both the theoretical and empirical implications of our findings.

**Supplementary Materials:** The following are available online at http://www.mdpi.com/2226-471X/4/4/78/s1, Dutch Grammatical Gender, Table S1: Participant group characteristics from the Language History Questionnaire.

**Author Contributions:** Conceptualization, J.R.V.K., F.R. and M.C.P.C.; Methodology, J.R.V.K., F.R. and M.C.P.C.; Software, J.R.V.K., F.R. and M.C.P.C.; Validation, J.R.V.K., F.R. and M.C.P.C.; Formal Analysis, J.R.V.K.; Investigation, F.R.; Resources, F.R. and M.C.P.C.; Data Curation, F.R.; Writing-Original Draft Preparation, J.R.V.K., F.R. and M.C.P.C.; Writing-Review & Editing, J.R.V.K., F.R. and M.C.P.C.; Visualization, J.R.V.K.; Supervision, M.C.P.C.; Project Administration, J.R.V.K., F.R. and M.C.P.C.; Funding Acquisition, N/A.

**Conflicts of Interest:** The authors declare no conflict of interest.

## Appendix A

**Table A1.** Experimental materials used in the study.

| Stimuli | Condition | Translation | Gender |
|---|---|---|---|
| Mi amigunan tin un banjo *koró* [1]. | Adj | My friends have a red bathroom. | M |
| E kamisa tin hopi boton *koró*. | Adj | The shirt has many red buttons. | M |
| E mucha a disidí ku nos mester traha *una* pisina *rondá* di e buraku. | Det + Adj | The child decided that we had to make a round swimming pool out of the well. | F |
| *El* kurason a kuminsá bati masha lihé. | Det | The heart started beating faster. | M |
| *El* ehérsito semper tabata *armó*. | Det + Adj | The army was always armed. | M |
| *Las* islanan ta *chikita*. | Det + Adj | The islands are small. | F |
| *Las* bentananan habrí ta bunita. | Det | The open windows are beautiful. | F |
| *Las* bòternan ta será ku un tapa temporal. | Det | The bottles are closed with a temporary cork. | F |
| Mi primu semper ta bebe biña *koró*. | Adj | My cousin always drinks red wine. | M |
| Kada djadumingu nos ta bai na misa *blanka*. | Adj | Every Sunday we go to the white church. | F |
| E stranheronan ta bebe serbesnan bon *fria*. | Adj | The foreigners are drinking nice cold beers. | F |
| *Las* palombanan *preta* ta kome pan. | Det + Adj | The black pigeons are eating bread. | F |
| Einan mester a bende *los* piskánan. | Det | They sold the fish. | M |
| E mucha hòmber ta bisti un sombré *chikito*. | Adj | The boy is wearing a small hat. | M |
| *Las* baiskelnan tin *una* kadena korá. | Det | The bicycles have a red chain. | F |
| *Los* brasanan di e señora ta *blanko*. | Det + Adj | The arms of the woman are white. | M |
| *La* pluma *blanka* ta suave. | Det + Adj | The white feather is soft. | M |
| *Los* kangreunan ta kome e piedra. | Det | The crabs are eating the stone. | M |
| E eksibishon tin *una* pintura *preta*. | Det + Adj | The exhibition has a black painting. | F |
| Mi ta stima *el* aros *blanko*. | Det + Adj | I love the white rice. | M |
| *Los* paranan *chikito* ta kanta *bunito*. | Det + Adj | The small birds are singing beautifully. | M |
| Mi amigu tin *una* kara *rondá*. | Det + Adj | My friend has a round face. | F |
| *Los* pannan *preto* no ta dushi. | Det + Adj | The black loaves are not delicious. | M |
| Mi bisiña tin un kabai *preto*. | Adj | My neighbor has a black horse. | M |
| E hembra ta yuda brui *los* webunan. | Det | The female helps breading the eggs. | M |
| Den *la* kaha korá, e hòmber a haña algun potrèt. | Det + Adj | The man found a picture in the red box. | F |
| *Una* bela *blanka* a paga durante e seremonia. | Det + Adj | The white candle went out during the ceremony. | F |
| Mi tin un mapa *koró* di mundu. | Adj | I have a red world map. | M |
| *Las* uñanan di mi bisiña ta *preta*. | Det + Adj | The nails of my neighbor are black. | F |

**Table A1.** *Cont.*

| Stimuli | Condition | Translation | Gender |
|---|---|---|---|
| Mi a kumpra *una* mesa *rondá*. | Det + Adj | I bought a round table. | F |
| *Los* sapatunan tin furu *preto*. | Det + Adj | The shoes have black lining. | M |
| Nos ta respetá *las* banderanan komo un símbolo nashonal. | Det | We respect the flags as a national symbol. | F |
| *Las* kamisanan *blanka* ta grandi. | Det + Adj | The white shirts are large. | F |
| Kòrsou tin hopi playa *turística* | Adj | Curaçao has many touristic beaches. | F |
| Mi ruman tin un kurpa *delegó*. | Adj | My brother has a skinny body. | M |
| *Los* avionnan ta *chikito*. | Det + Adj | The airplanes are small. | M |
| Mi ofisina ta un edifisio *koró*. | Adj | My office is a red building. | M |
| Ayera mi a kumpra kuminda *spañá*. | Adj | Yesterday I bought Spanish food. | F |

[1] Items in italics were experimentally manipulated to exhibit Spanish-like gender agreement. Det = Determiner, Adj = Adjective, M = masculine, F = Feminine.

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
