# Peer review of "Spanish Grammatical Gender Interference in Papiamentu"

_languages, doi:10.3390/languages4040078_

Round 1
Reviewer 1 Report
The paper refers to HL and to L2. I would thus suggest to refer to L1 rather than to mother tongue., as it is done today all over. On P2, L58 what is meant by 10 traditional speakers of Palenquero? If it just means L1 monolingual speakers of Palenquero, say so, please, and erase traditional.
P2, L76 Please, make explicit what you mean when you say that the fact that L1 speakers of Palenquero changed nouns ending in –a to Palenquero invariable –o, whereas L2 speakers never did, demonstrates Spanish interference.
P4, L128-130 Please clarify: which one is the HL, Spanish or Papiamentu, and why?
The author is rather selective in describing the various genders systems, as i.e. the Spanish and the Papiamentu systems are described, but the Dutch one is not. It would help the reader to describe all the gender systems treated in the paper.
In general, the consecutio temporum should be respected, as e.g. P4, L122-128 contain: lived … took place … are exposed … consists … were born … speak Spanish, etc., with a lot of mixing between past tenses and presents.
General comments
The paper refers to HL and to L2. I would thus suggest to refer to L1 rather than to mother tongue., as it is done today all over.
On P2, L58 what is meant by 10 traditional speakers of Palenquero? If it just means L1 monolingual speakers of Palenquero, say so, please, and erase traditional.
P2, L70-71 Same comment for 12 traditional Palenquero speakers,
P2, L75 same for traditional speakers
P2, L76 Please, make explicit what you mean when you say that the fact that L1 speakers of Palenquero changed nouns ending in –a to Palenquero invariable –o, whereas L2 speakers never did, demonstrates Spanish interference.
P2, L79 What are the consequences for the teachers of Palenquero that this is not their dominant language?
P3, L90-92 Please, clarify.
Review of Spanish grammatical gender interference in Papiamentu
P4, L118-130 The presentation of the various groups is somehow chaotic.
P4, L128-130 Please clarify: which one is the HL, Spanish or Papiamentu, and why?
P4, Table 1 and P5, Table 2 The tables should be reorganized in order to be better readable.
Specific comments
The author is rather selective in describing the various genders systems, as i.e. the Spanish and the Papiamentu systems are described, but the Dutch one is not. It would help the reader to describe all the gender systems treated in the paper.
In general, the consecutio temporum should be respected, as e.g. P4, L122-128 contain: lived … took place … are exposed … consists … were born … speak Spanish, etc., with a lot of mixing between past tenses and presents.
P(AGE) 1, L(INE)23 The speaker must suppress gender in their L2 à Speakers must suppress gender in their L2 OR The speaker must suppress gender in his/her L2
P1, L38-39 often speaking to varying degrees Papiamentu, Dutch, English and Spanish. à often speaking Papiamentu, Dutch, English and Spanish to varying degrees.
P2, L42 Spanish nouns are either grammatically masculine or feminine à Grammatically speaking, Spanish nouns are either masculine or feminine.
P2, L70 whether they had found it to be acceptable à whether they had found it to be acceptable or not.
P3, L95 test if à test whether
P4, L124 traditional Papiamentu speakers à L1 Papiamentu speakers
L128 as an adult à as adults
P5, L176 Please explain what are some of the properties of the ‘omnibus’ model.
P10, L311 (Munarriz ete al. 2019) à (Munarriz et al. 2019)
Author Response
REVIEWER 1
Overall, I enjoy this article. It is generally well written and easy to follow, with a clear
goal and outcomes. My biggest concern is that there seems to be more clarity and detail
that needs be provided in almost all sections. I organize my comments by section, without
differentiating between the small issues (e.g., typographical) and larger ones.
We thank you for your helpful comments!
MINOR COMMENTS PER SECTION:
Introduction 30 – It’s not clear what is meant by “may become exaggerated”. Can you please clarify?
We mean that cross-linguistic transfer is more likely to occur in the case of highly cognate languages with grammatical differences due to increased lexical and semantic overlap between the two languages. We have added additional clarification to the sentence.
46 – APA style states that abbreviations like i.e. should be in parentheses, and when
wanting to express that outside of parentheses, the full lexical equivalent is preferred.
Parentheses have been added throughout the document.
56 – The same for e.g.
Parentheses have been added throughout the document.
70 – It’s not clear what is meant by “whether they had found it to be acceptable.” Originally, I understood this to mean “regardless of whether they found it acceptable or not (they had to repeat the sentence as is)”. But perhaps it means they had to repeat their acceptability rating? Or something else entirely perhaps? Please re-word to make the meaning obvious.
They had to repeat every sentence exactly as they heard it, “good” Palenquero or not. If they found the sentence not to be acceptable or “bad” Palenquero, they still had to repeat each sentence as they heard it. We have rewritten the sentence for clarity.
86 – It is understandable to not list all the details of what the exact experimental tasks
Lipski (2017) used if there are several, but perhaps after “experiments” there could at least
be a parenthetical clarification of what kind(s) of on-line tasks were included.
The experiments that he carried out were: (1) Number Recall + Repetition and (2) Speeded Translation. We have now added this information.
86-87 – Saying “the strength of Spanish gender agreement among L2 Palenquero
speakers” makes it sound like the gender agreeing is happening in Spanish (as opposed to
Palenquero). I would clarify by saying something like: “the strong influence of Spanish
gender agreement on the Palenquero of L2 speakers.”
We have added the reviewer’s suggestion to the sentence.
88 – I’m not sure “interestingly” is the best transition here, as it really is not particularly
surprising the heritage speakers are in the middle given how they patterned half the time
with the L2 speakers and half the time not in Lipski (2015).
“Interestingly” was changed to “Furthermore”.
94-104 – For me, this is the beginning of where the description of the participants starts to be confusing. You mention earlier that Papiamentu speakers in Curacao are highly multilingual (line 38), so it is not immediately clear what is meant by these groups. You need to be as explicit as possible. For example, with a term like Spanish heritage speaker, if Spanish is their heritage language, what is their other (dominant) language? And are they bilingual, trilingual, or more? Also, you use the term traditional Papiamentu speaker here, but later refer to them as L1 Papiamentu speakers. I would be as clear as possible about who these participants are from the onset, and then be as consistent as possible?
We have added footnote 1 to describe why the participants are referred to as multilingual while also providing some clarifying language about the characteristics of each participant group. We have also provided a new Table S1 in Supplementary Materials that provides further details from the LHQ on each of the participant groups, and we have changed the group names to:
Dutch Dominant (N = 7)
Papiamentu Dominant (N = 22)
Spanish Dominant (N = 6)
Heritage Spanish Papiamentu (HS) (N = 6)
104 – “Greater” than what?
We have deleted “greater”. The sentence is now as follows: “…. in order to test whether the presence of grammatical gender more generally, may lead to gender agreement interference.”
113 – Because you are including Dutch-Papiamentu speakers, you need to also include a
Description of how gender works in Dutch. Otherwise, making a claim that the gender of
Dutch can also interfere lacks background information about how that could be possible.
We have added additional information on the Dutch grammatical gender system on lines 106-114 and have included extended information in Supplementary Materials.
Materials and Methods
117 – Following up on the comment above about defining the participant groups clearly,
because two of the group names have only L1 in the title, it could be easily misinterpreted
that one or both of them are monolinguals. Perhaps it would be worthwhile to state from
the onset that all groups are (to at least some extent) bilingual. And/or the groups could be
re-named, for example: Dutch-Papiamentu, L1-Papiamentu L2-Spanish, L1-Spanish L2-
Papiamentu, Heritage Spanish-Papiamentu.
We have renamed the groups to make clearer that all participant groups are multilingual, now making reference to their language dominance or heritage language status.
117-121 – The description of the different groups is currently quite limited, and it is hard
to truly compare the similarities and differences among them. It would be helpful to flesh
out this information more in the form of a table, because not all groups are defined by the
variables listed. The Spanish heritage speaker group is particularly underdefined.
Essentially, by expanding (3) and making a column for each of the details listed after it,
readers would have a nice, clear overview of the different groups. This would also help
the afore mentioned issue about being monolingual/bilingual. It would also help clear up
the earlier statement about the population being highly multilingual in the four languages.
Due to space limitations, we have added additional group characteristics from their language history questionnaire as Table S1 in the Supplementary Materials.
121 – If the Dutch-Papiamentu group was born and raised in Curacao, is L2 Papiamentu
is best way to qualify them (as they are described in line 102)? Even if Dutch is their
dominant language and the one spoken at home, L2 doesn’t seem entirely accurate. Why
not just refer to them as Dutch-dominant and set aside the labeling of L1 or L2? Or if there
is more explanation/clarification why L2 is the best way to describe them, please include it.
The language status and use of Dutch and Papiamentu on the islands is complicated and often intersects with race (see Kester 2011). This leads to situations in which despite Papiamentu competency and instruction in schools, Dutch speakers descended from (mostly white) Dutch families often do not speak Papiamentu outside of school settings. We have changed the name of this group to “Dutch Dominant”.
132 – “Eighty-two Papiamentu sentences were created, of which 40 contained a
Spanishlike…”
The formulation of this sentence has been changed accordingly
133 – Later in the results, it is clear that there are three conditions about where the gender interference may be occurring (adjective, determiner, or both), but you should be clear about that here when describing the stimuli that there are three defined conditions. What percentage of the stimuli had both, what percentage had just the determiner, and what percentage had just the adjective?
We have now specified the three condition types at first mention of the experimental stimuli, and we have included the number of experimental stimuli per condition.
135 – “ending in -o” only applies to the adjective, as the determine shows masculine
gender agreement differently.
This sentence has been changed/specified: “(e.g. adjectives ending in –o)”.
137 – What was verified? Please clarify for the reader.
The sentence has been changed to: “Two native Papiamentu speakers from Curaçao verified if the manipulated stimuli were correct Papiamentu sentences, apart from the experimentally manipulated target determiners and/or adjectives.”
138 – What was the intention of editing? Please clarify for the reader.
The correct target items had to be cut out and all of them had to be the same length. A few targets had to be recorded multiple times and the right ones had to be cut out and put in the experiment. We have added some clarifying language to the sentence.
142 – “…used in the experiment, containing a masculine…”
143 – Although the intended meaning is understood, I would avoid using “correct” here when providing the stimuli. “Papiamentu equivalent” would be less problematic.
This has been changed to “Papiamentu Equivalent”
143 – The adjective provided in (5b) is an adverbial adjective, which has gender in
Spanish, but it is always masculine when it does not have the morphological suffix -mente
added (which requires feminine -a-). That means that although the example provided is
still an example of Spanish-like masculine gender because it switches from the Papiamentu -a to -o, it is not technically an example of gender agreement because the adjective would always be masculine regardless of the subject. What percentage of the stimuli were like this? It is possible that the gender interference of adverbial adjectives and traditional agreement adjectives are different, and they should be separated in the data. Relatedly, this sentence has two manipulated adjectives in one. Did all the adjective-only stimuli have such a setup or only some? I would re-work Tables 1 and 2 to be an overview of all the manipulations together so that the combinations can be understood clearly by the reader.
We thank the reviewer for their careful eye. The reviewer is correct, and we have acknowledged this in footnote 4. This is the only sentence with this characteristic, and it also includes a nominal phrase with the “correct” experimental manipulation on the adjective, so we kept the sentence in the analyses. We subsequently went over all materials again and caught two additional sentences that were miscoded. These had no other “correct” manipulations, so they were removed from all analyses (which did not change any of the significant results). We included footnote 2 that acknowledges this coding mistake. The list of experimental sentences is included in Appendix A.
144-145 – Were these positions the only two options (i.e., postnominal and postverbal adverb)? Then describe it as such or include the full list of positions. Also, was the cognate noun always the subject? Be explicit. More generally, an appendix with all the stimuli, including filler sentences, would also be helpful.
The adjectives were either postnominal or predicate adjectives, which we now mention in the text. The target noun was not always the subject. We now include a list of experimental sentences in the Appendix.
160 – Is there a reason the instructions were only provided in these languages? Was the language of instructions decided by group or by individual participant? Wouldn’t Papiamentu have been the best choice since that is what the task is? Or why not have all four as options?
The language of instruction was provided in the languages that the second author, who tested all participants, could comfortably provide them in. We have added more information in the sentence.
163 – Does this include the Dutch-Papiamentu group (meaning this group only had the option to complete the forms in Papiamentu)? Again, why not have Papiamentu be the only option? Or why not all four?
All groups had the option to fill out forms in Spanish and Papiamentu. With the exception of the L1 Spanish (now Spanish-dominant) group, the remaining three groups had formal instruction in Papiamentu and were literate in Papiamentu.
Results
165 – Define "accuracy.” In other words, state explicitly how the “yes” and “no” responses were scored and calculated.
We have now defined accuracy in this section for both experimental and filler trials.
168 – It is unclear what “correct” means here as the filler trials were explained to be “correct” previously (in line 136). Does that mean participants had to say “yes” to all filler sentences for them to be counted as “correct”? Or were there ungrammatical filler sentences? Either way, you should provide examples in the methods.
We have now defined accuracy in this section for both experimental and filler trials.
173-176 – You should state explicitly that this analysis only included the target stimuli (not fillers). You should also define the independent variable, as it not necessary the same as “accurate/correct” for the prior filler items. In other words, state explicitly how the “yes” and “no” responses were scored and calculated for the target stimuli.
We have now clarified that the dependent variable is on the proportion accuracy score for experimental items (in which participants correctly identify experimental trials as not well-formed Papiamentu sentences, i.e. correctly responding ‘no’).
181-182 – Thus far this group has been referred to as the Dutch-Papiamentu group, so it should be consistent in both the title and the first sentence of this section.
We have now made group names consistent throughout the paper.
184-185 – It is not clear why pointing out that this group descriptively was least accurate with the masculine determiners if that finding was not statistically significant. What was significant (and worth stating) was that they were overall less accurate with masculine. Also, was there a post-hoc analysis to determine the differences between conditions? Visually it looks like they were most accurate with the combined Det+Adj condition, but it is unclear if there is a difference between the other two.
We meant to describe the main effects for Condition and Gender. We have rewritten the sentence for clarity.
194-202 – It is hard to follow the pairwise comparisons here. Please be clearer in your explanation of the different comparisons that were run. My impression is that the description is too abbreviated, which causes confusion, requiring multiple re-reads.
We have added some clarifying language to better describe the comparisons tests (e.g., comparison with same condition type across gender, e.g. Masc Det v. Fem Det as well as across condition but same gender, e.g., Masc Det v Masc Adj).
198 – “Amongst contrasts…”
214-214 – Doesn’t this also mean they were most accurate at accepting the Spanish-like masculine-marked features when manipulated on the adjective?
Yes, we believe that this would be a fair characterization.
229 – A summary of the main findings for all groups would be helpful.
We include a general overview of basic findings in lines 258-269.
Discussion
235 – The phrasing “differ in the realization of gender features” seems odd here, as Spanish and Papiamentu don’t differ in how, Papiamentu just does not. The wording makes it sound like both have a realization of gender, but they happen to be different.
We have changed this sentence.
235-241 – I do not disagree with the general pattern presented here, but with the information provided in the methods about the participants, it doesn’t seem we can rule
out that it merely a proficiency effect. If the Papiamentu proficiency level for the four
groups is hierarchical with the L1 Spanish group being the least proficient, then DutchPapiamentu, then heritage, then Papiamentu L1, it would make perfect sense why the groups would differ in the performance on the task, as it is measuring their ability to identify grammatical errors. You should either acknowledge that proficiency could be a confounding variable or make more of an explanation about how it is not. For one, providing more of that information in the description of the participants (including the table) would be beneficial. Also, you could run a statistical analysis on the filler items, and if the groups perform uniformly, that would indicate that their proficiency level is equivalent enough in that regard. Relatedly, I think there is a lot more that could be said and expanded on regarding the role of dominance. However, that would also entail more detail about dominance in the description of the participants in the methods.
We agree with the reviewer’s critiques and we have added additional language acknowledging the role that proficiency may play as well as acknowledging our small sample sizes.
237 – I think you need to unpack the possible influence of Dutch gender more. It seems to be buried here within this paragraph. First, do the Dutch-Papiamentu speakers have any proficiency in Spanish and/or how knowledgeable are they of the gender system in Spanish? And is it similar at all to the gender system in Dutch? This is where tying back to a description of how gender works in Dutch would be helpful. The mere fact that both languages have gender is not enough to argue for such an influence. Why should the fact that a Dutch speaker would be less able to reject a Spanish-like masculine determiner (because of Dutch gender) in Papiamentu is currently opaque to the reader. Again, going back to all of these Papiamentu speakers being highly multilingual raises questions. Why
would it be that Dutch is interfering, when it could also be some level of Spanish interfering? Without better describing the full language profile of this group (and all groups), it is hard to make such claims.
We have now added more information on the Dutch grammatical gender system and point out parallels between Dutch and Spanish in the Discussion.
245 – I would not describe it as mixed exactly. There was only one instance where some participants were less accurate with feminine, and that was the L1 Spanish group’s adjective condition. To me that seems like an exception compared to how consistently masculine was the one more difficult anywhere else there was a difference. On that note, since feminine was easier, perhaps it would be worth tying in how masculine is assumed to be the default gender in dual-gender languages like Spanish. That is to say, the fact that availability of a feminine feature makes it easier to suppress that feature. Masculine, as a non-feature, is hard to suppress because the question could be, what are you suppressing?
We have reworded this paragraph along the reviewer’s suggestions and have added more information on the default-status of masculine in Spanish as a plausible explanation for the overall results.
249-267 – The discussion of the parallels with code-switching is interesting, however, it needs to be more explicitly tied to the current study.
With the slightly expanded discussion, we hope that the transition to finding parallels with prior work from our research groups is now clearer. We hope it is clear that this is a post-hoc observation.
273 – The paper needs to address that the low number of participants in all groups except the L1 Papiamentu is a limitation of the study. Any claims for the other groups need to be very suppositional. The pattern is promising, but the statistical analysis cannot be truly relied upon, and it likely could explain the mixed results.
We have acknowledged our low sample sizes and have attempted to indicate our desire to extend the study to a new group of Papiamentu speakers residing in The Netherlands.
273 – Overall, the discussion could use more connection to the broader field. Why are these findings important to the literature on acquisition? Why are these findings important to the literature on gender assignment?
We believe that further expansion of the discussion would exceed our word limit of 4000 words (we are already over) and will take this comment into account when submitting a fuller, more comprehensive study with more robust sample sizes.
Reviewer 2 Report
See my comments on the attached manuscript. In general I think this study and its results could be interesting but is currently not presented as being theoretically motivated, the results are not adequately compared to other studies except for Lipski (these are clearly the inspiration for the current study, but should not be the only ones discussed), and it is not discussed whether these results are what we would expect or not and why. The Lit Review, Discussion, and Conclusion need to be further fleshed out in order for this work to be appropriately contextualized and its significance appreciated.

Author Response
REVIEWER 2
The paper refers to HL and to L2. I would thus suggest to refer to L1rather than to mother tongue., as it is done today all over. On P2, L58 what is meant by 10 traditional speakers of Palenquero? If it just means L1 monolingual speakers of Palenquero, say so, please, and erase traditional.
The language adopted was taken directly from Lipski’s studies, however, we have changed the label to L1. Lipski does not clarify on whether they are monolingual speakers.
P2, L76 Please, make explicit what you mean when you say that the fact that L1 speakers of Palenquero changed nouns ending in –a to Palenquero invariable –o, whereas L2 speakers never did, demonstrates Spanish interference.
For this experiment, all participants were asked to repeat each sentence exactly as they had heard it (meaning that they had to repeat the words that contained feminine gender agreement as well). However, despite these instructions, many L1 Palenquero speakers ‘corrected’ these sentence to the Palenquero invariable –o (i.e., a correction of the manipulated Spanish-like form). On the other hand, L2 Palenquero speakers seemed to accept the repetition of Spanish-like feminine gender agreement in the Palenquero sentences. We have added some clarifying language to this section.
P4, L128-130 Please clarify: which one is the HL, Spanish or Papiamentu, and why?
Spanish is the heritage language. We have changed the group name to Spanish HS-Papiamentu to help clarify this.
The author is rather selective in describing the various genders systems, as i.e. the Spanish and the Papiamentu systems are described, but the Dutch one is not. It would help the reader to describe all the gender systems treated in the paper.
We have added some information on Dutch grammatical gender on lines 109-119 as well as a greater description as Supplementary Materials.
In general, the consecutio temporum should be respected, as e.g. P4, L122-128 contain: lived … took place … are exposed … consists … were born … speak Spanish, etc., with a lot of mixing between past tenses and presents.
We thank the reviewer for their editing comment and have made changes throughout the manuscript.
General comments
P2, L70-71 Same comment for 12 traditional Palenquero speakers,
We have changed this to L1.
P2, L75 same fortraditional speakers
We have changed this to L1.
P2, L79 What are the consequences for the teachers of Palenquero that this is not their dominant language?
The broader point is that their profession as language instructors probably suggests that they have greater language control in suppressing Spanish interference (what is meant by metalinguistically sensitive).
P3, L90-92 Please, clarify.
Review of Spanish grammatical gender interference in Papiamentu
We have changed the subheading.
P4, L118-130 The presentation of the various groups is somehow chaotic.
We have labeled the groups making reference to language dominance and provided additional information on the groups in text as well as part of Supplementary Materials.
P4, Table 1 and P5, Table 2 The tables should be reorganized in order to be better readable.
We have reorganized the tables and hope to have achieved greater clarity.
Specific comments
P(AGE) 1, L(INE)23 The speaker must suppress gender in their L2 à Speakers must suppress gender in their L2 OR The speaker must suppress gender in his/her L2
The sentence has been changed to ‘this suggests that in cases where the speakers must suppress gender in their L2’
P1, L38-39 often speaking to varying degrees Papiamentu, Dutch, English and Spanish. à often speaking Papiamentu, Dutch, English and Spanish to varying degrees.
The sentence has been changed to ‘Papiamentu speakers in Curaçao are highly multilingual, often speaking Papiamentu, Dutch, English, and Spanish to varying degrees’
P2, L42 Spanish nouns are either grammatically masculine or feminine à Grammatically speaking, Spanish nouns are either masculine or feminine.
The sentence has been changed to ‘Grammatically speaking, Spanish nouns are either masculine or feminine’
P2, L70 whether they had found it to be acceptable à whether they had found it to be acceptable or not.
The sentence has been changed to ‘repeat each sentence exactly as they had heard it, regardless of whether they had found it to be acceptable or not’
P3, L95 test if à test whether
The entire sentence has been formulated differently
P4, L124 traditional Papiamentu speakers à L1 Papiamentu speakers
We have changed this.
L128 as an adult à as adults
This has been changed to ‘as adults’.
P5, L176 Please explain what are some of the properties of the ‘omnibus’ model.
The omnibus model is explained on lines 195-200. Here, omnibus just refers to the highest order statistical model which includes all groups.
P10, L311 (Munarriz ete al. 2019) à (Munarriz et al. 2019)
This has been corrected.
Reviewer 3 Report
See attached.

Author Response
REVIEWER 3
Abstract and Introduction
17 – “Spanish Heritage Speakers, and Spanish L1 Speakers”: did these speakers speak Papiamentu too?
Yes, they did.
45 – “and” à but
This has been changed
70 – ‘Acceptable or not’
Sentence has been changed to ‘regardless of whether they had found it to be acceptable or not’
General comment P3 – So far what I’m missing is a theoretical background on the acquisition of syntax and/or language contact effects on syntax. What does theory predict that should happen with gender? Is it more or less susceptible to change than other syntactic elements?
We appreciate this comment but do not believe that we can adequately address this comment (especially due to our word limit constraints). We have attempted to interpret the results making reference to masculine as default, which could explain the gender asymmetry that we generally found.
107 - “First Lipski’s participants were dominant Spanish speakers acquiring L2 Palenquero…” à Not all of them, per lines 58 + 59
The sentence has been changed to ‘First, most participants in Lipski’s experiments were dominant Spanish speakers acquiring L2 Palenquero..’
130 – Where do these speakers live now? When did they move?
‘All participants of this group live in Curacao today’ has been added.
Materials and Methods
132 - Explain/clarify this – these words were inserted in Spanish?
Stimuli were meant to appear more Spanish-like. We would not characterize them as having been inserted in Spanish.
133 until 135 – It was unclear until you talked about the “fillers” that these 40 sentences had been manipulated
We have added ‘manipulated’ to the beginning of this section in order to make this clear from the beginning
137 – “verified”: what does this mean? Since they clearly were no longer grammatical by design
The sentence has been changed to ‘Two native Papiamentu speakers from Curaçao verified if the manipulated stimuli were correct Papiamentu sentences, apart from the experimentally manipulated target determiners and/or adjectives’.
143 example 5b – “kanta bonito”: you said in line 47 that a Papiamentu adjective can end in –o. Did all your examples require –u, or were there some correct in –o as well? Where those what your 42 filler sentences were?
Due to the comments from Reviewer 1, we have recognized in Footnote 4 that this specific item should not have been modified due to its adverbial use. However, we kept this sentence in because it contained a correctly manipulated noun phrase in the subject. We now include a list of experimental stimuli in the appendix. Filler items were never modified to appear Spanish-like. Some Papiamentu adjectives are correct in –o and have been used in the experiment. For example the Papiamentu adjective ‘rondó’ (correct form) has been changed to the feminine adjective ‘pisina rondá’
151 - Why not the plurals un@s though?
The plurals unos/unas could have been used as well. Instead, we used the plural forms los/las
154 general comment: # of tokens overall? # exluded?
We have added that 1558 tokens were analyzed out of a possible 1640 on line 204.
Discussion 231 – Include these in your literature review
We have now made reference to difficulty in gender agreement in L2 speakers on lines 34-37. We are aware that the treatment is light but cannot see how to adequately treat this issue within our 4000 word limit.
242/243 – Why might this be? (more interference with Determiners)
We have added some possible accounts for the difference on lines 279-282.
247: Explain this in the Literature Review
We have introduced this information on lines 45-47.
249 until 251 – None of this was in the literature review
The study was not conceptualized or designed to test code-switching, so we do not believe it warrants being included in the literature review. We are providing a post-hoc connection to code-switching based on the results that were obtained.
252 – “the analogical criterion strategy”: explain
We have added some clarifying information after first mention.
Round 2
Reviewer 3 Report
The authors have responded well to the comments provided. And it now includes the clarity and detail I was hoping for. The only necessary edits I see now are minor copy edits (e.g., in line 305 'The' shouldn't be capitalized, and so on.).